# Social Risks of International Labour Migration in the Context of Global Challenges

**Aleksandra Kuzior [1,***], Anna Liakisheva [2], Iryna Denysiuk [3], Halyna Oliinyk [4] and Liudmyla Honchar [5]**

[1] Faculty of Organization and Management, Silesian University of Technology, 41-800 Zabrze, Poland
[2] Department of Social Work and Pedagogy of Higher Education, Lesya Ukrainka Eastern European National University, 43025 Lutsk, Ukraine; liakisheva@i.ua
[3] Department of Practical Linguistics, Pavlo Tychyna State Pedagogical University, 20300 Uman, Ukraine; ira_denysiuk@i.ua
[4] Department of Social Pedagogy and Social Work, Ternopil Volodymyr Hnatiuk National Pedagogical University, 46027 Ternopil, Ukraine; galochka_o@ukr.net
[5] Laboratory of Institutional Education, Institute of Problems on Education of the National Academy of Educational Sciences of Ukraine, 04060 Kyiv, Ukraine; honchar_luda@ukr.net
* Correspondence: lubanes@gmail.com

**Abstract:** The results of the study of migration risks of labor migrants from Ukraine are presented in this article. The purpose of the study is to find out the differences in the perception of obstacles and risks that arise in the process of work abroad among experienced and potential labour migrants from Ukraine within the cognitive, behavioural, and emotional components of their intercultural competence. The study has been implemented from the standpoint of a set of analytical tools, including: the concept of the advantages of replacing the "risk/reliability" scheme with the "risk/hazard" scheme; views of risk and chance as interrelated variables that motivate people to try to explore the world and overcome obstacles; the concept of "triple individualization" in a risk society. It has been found that social risks are hidden in the imbalance of intercultural competence of experienced labor migrants and are not realized by potential labor migrants. It has been proven that the greatest social danger for labor migrants from Ukraine is the loss of components of competence and initiative. It has been established that the key points of the comparative analysis of social risks faced by labor migrants from Ukraine open up prospects for improving the methodology for studying social (and socio-cultural, in particular) risks.

**Keywords:** social risks; labour migration; social insecurity; cross-cultural competence; migration flows

## 1. Introduction

Current global trends create preconditions for the in-depth scientific understanding of international migration and associated social risks, both in migrant origin and host societies. The intensification of migration processes poses risks of an economic, socio-cultural, and political-managerial nature. In the economic sphere, a significant share of migrants on the labour market leads to increased unemployment among host countries, lowering of the minimum wage, and a spread of shadow labour relations. The socio-cultural and political management faces challenges related to integration of migrants into host societies and associated problems (growth of xenophobic sentiments, social exclusion, ghettoization of migrants, etc.) (Golobof et al. 2011).

Migration flows donor countries also face the multidimensional effects of external migration, with the most notable being a reduction of labour resources, namely the loss of professional skills by

a significant proportion of migrant workers who have to practice a profession they had not been trained for, their precarization, and demographic problems, etc. (Lindert et al. 2009). Functional implications for donor countries are also numerous, e.g., mitigation of unemployment, cash inflows that stimulate production, improvement of economic dynamics, etc. (Li et al. 2017).

A number of scientific schools have dealt with the problems of labour migration. Thus, representatives of the school of human capital (Chun and Shin 2018; Rodrigues 2018; Sanders et al. 2018; Wu et al. 2018; Zhang et al. 2018) consider the causes of labour migration from the standpoint of the competence approach.

International migration processes have become not only an important factor of economic and cultural mutual enrichment, but also a factor in the spreading of all kinds of challenges, which necessitates a comprehensive study of migration risks, in particular from a sociological perspective. Thus, the practical importance of the study is determined by both the socio-political processes of the present and the institutional needs of governmental and international organizations involved in labour migration regulation.

It should be noted that there has been a significant revival of scientific interest in the issue of migration risks in recent years. The role of risk theory, formulated in the late 1960s and early 1970s, is important in the study of international migration. Studying social risks of labour migration requires deepening both the methodological foundations of relevant researches and refining the conceptual framework. It is advisable to distinguish between the socio-cultural, socio-economic, socio-legal, and socio-political spheres of actualization of migration risks that may occur at micro, meso, and macro levels. Socio-economic risks are associated with the risk of socio-economic destabilization of donor and recipient societies of migratory flows, with the possible loss of socio-professional status, loss of professional skills, lowering quality of life for migrants, etc. Socio-cultural risks derive from differences in value systems, practices, and behavioural models. Migration security risks relate to the impact of migration processes on crime rates, and the objective and subjective security of migrant workers and residents of host societies.

It should be noted that the relevance of the study in terms of theory stems from lack of thorough theoretical studies and tools for the empirical research of migration risks arising in the course of labour activity and cross-cultural interaction of migrants abroad.

In recent years, there has been an increasing number of publications on risk in all areas of human activity (Fromhold-Eisebith 2018). This trend has led some researchers to talk about the formation of an integrative discipline, namely risk studies (Pécoud 2018; Vo and Zhang 2019).

Currently, risk studies is an interdisciplinary field of knowledge, the subject of which is natural (Gheasi and Nijkamp 2017), technogenic (Hilorme et al. 2020), and sociogenic risks (Svetlana Nesterenko et al. 2019).

A methodology for the study of theoretical and practical aspects of international labour migration has been elaborated in scientific works.

Thus, the application of the phenomenological approach involves determining the feasibility of negative scenarios, based on the analysis of necessary and sufficient conditions associated with the laws of nature (Barsbai et al. 2017). The method can be implemented on the basis of fundamental laws, which makes it possible to determine the state of individual components of the studied system. A significant drawback is the unreliability of the results in the study of transients, as well as the inadmissibility to assess the risks associated with the functional reliability of individual components of the object of analysis. These shortcomings make this method unacceptable for assessing social risks.

Cost-effectiveness analysis involves comparing relative forecast costs for a number of possible risk management options, and they all serve to achieve a common goal (Hynie 2018). At the initial stage of the analysis, a detailed description of the goal is defined and carried out, after which the process of implementing measures to reduce a specific type of risk is detailed. The number of possible options depends on the nature of the specific problem, but in a typical cost-effectiveness analysis, 3–5 different strategies are usually considered. This method aims to identify the most cost-effective ways to achieve

the desired management result. The advantages of the method of cost-effectiveness analysis are that, in its application, the calculations are direct. This makes it possible to determine the least costly management option from a number of possible options, having a minimum amount of information. The disadvantage of the method of cost-effectiveness analysis is that it is somewhat limited in scope, the stated task is not always considered desirable after completion of the study. In addition, the quality of different management options can be assessed only by their cost indicators. It is equally important that the cost-effectiveness analysis is not able to values seems quite problematic and more research is needed to solve this problem.

The cost–benefit analysis involves comparing the estimated costs of improved risk management and the expected benefits of reducing the risks associated with different scenarios for solving existing problems (Kuzior et al. 2019). In cases where, in monetary terms, the benefit–cost ratio is greater than one, this means that the expected benefit of the measure is likely to outweigh the management costs. On the other hand, if the benefit-cost ratio is less than the allowable value, it indicates that the chosen strategy is economically unjustified. The use of the method is expedient in the process of making managerial decisions, as well as for the substantiation of the recommendations for changing the instruments and public policy measures.

The conceptual basis of the analysis of social risks is their study on the basis of the classification of areas of possible manifestation: the actual object of risk; object of risk in the system of socio-economic relations; attitude to the risk of the individual and society. Accordingly, the analysis of social risks can be conducted in terms of:

- Frequency and probability of adverse events or adverse effects (Chang et al. 2018;)
- Social interpretation of negative consequences taking into account social values and interests (Makedon et al. 2019).
- Possible material damage (Mou et al. 2018).
- Perception of risk by a person, depending on his (her) personal preferences, rather than their objective level (Czaika and Parsons 2017; Lissoni 2018; Kwilinski and Kuzior 2019).

It is revealed that international labour migration is the object of study of a number of disciplines, as well as interdisciplinary studies, in which the approaches of all scientific disciplines are equisignificant and complementary (Zhong et al. 2017).

From the standpoint of sociology, international migration is regarded as a social process, based on a system of factors of socio-economic, socio-political, and socio-cultural nature, which are prerequisite for relocation and adaptation in the host country (Rianne Dekker et al. 2018).

Distinction between repulsion and attraction factors that determines migration behavior is common in scientific discourse (Smith 2017; Tkachenko et al. 2019).

There are different opinions on the definition of these factors (Tetiana et al. 2019). It is determined that, in case of an inconsistency of internal (needs, interests, values of the population) and external factors (incentives offered in the field of labour activity), there are motives for the movement of the able-bodied population in the territorial space in the interests of a better satisfaction of actual needs, in particular, earning more income from work, avoiding the risk of poverty, ensuring the development of abilities, knowledge, and skills.

The spread of migratory sentiments is produced by such disincentives as conflict of social and labour relations, unfavorable working conditions and payment, limited social protection, lack of decent jobs, corruption in employment, lagging behind developed countries in terms of income, human development index, and also life threatening circumstances.

The study and analysis of the interaction of various internal and external motivational factors should be an important prerequisite for the development of preventive measures to effectively regulate labour migration at the stage of migration attitudes through increasing incentives and reducing anti-incentives in employment, living conditions, and population development.

The significance of changes related to relocation, weakening (or loss) of former social ties, and the need to absorb new cultural norms and integrate with new social groups makes it possible to consider international labour migration as a vital strategy (Yamori 2019). The vital strategy is determined by the survival of labour migrants in the recipient countries, while forming strategies for achieving the goals of labour behavior.

The relocation of international migrants is a way of realizing their choice, and the movement across administrative and cultural borders is an indicator that confirms the fact of migration (Light et al. 2017; Lyons 2017).

The specifics of the development of modern society are such that social reality is characterized by a significant dynamism of all processes and their uncertainty. Under such conditions, risk accompanies any purposeful activity of a social subject, in turn, the latter is aimed at reducing the uncertainty of its results.

The structure of social representation includes the central core and peripheral system. The central core is connected by the collective memory and history of the group, it is stable and performs the function of developing a social representation. The peripheral system ensures the integration of the individual experience of each member of the group, supports its heterogeneity, it is characterized by changes, contradictions, and adaptability to reality. The system allows the differentiation of content but protects the central core from external influences (Orbeta et al. 2009).

Based on the objectivity of social life, it is necessary to recognize the attributiveness of risk for the progressive development of society. Risk, in most cases, has a social nature since it is produced by social subjects and its actualization significantly affects their characteristics and interaction. Therefore, the concept of social risk is gradually acquiring the status of a general scientific category, in connection with which there is an expansion of the range of theoretical problems associated with the need to study its latest aspects (Bonoli 2005).

At the same time, the existing theories of social risk are characterized by significant imbalances between wide applied and insufficient methodological achievements (Schmidt-Soltau 2003) Thus, the need to expand the practice of including the human factor in risk analysis leads to the improvement of the methodological foundations of the analysis of the social aspects of risk, as well as the development of new management decisions aimed at minimizing the negative consequences of their implementation.

Krohn and Weyer (1994) propose to consider technical innovations and their inherent risks not only as the application of natural science practices, but also social ones. Since the latter are "hypothetical social structures", then, on the one hand, these structures must adapt to innovation. On the other hand, the introduction of innovative technologies can lead to structural social changes that are not always positive. Therefore, the processes of introducing technological innovations into social structures should be interpreted as attempts to "social implementation of inventions" that take place not in the labouratory but within the framework of sociotechnical activity, which is included in the professional, public, and even private social sphere (Olivier Pintelon et al. 2013).

Social risks in the context of ensuring national security are also considered by Taylor-Gooby (2004a). In particular, the scientist identifies migration risks, the main characteristics of which are the disintegration of the economic, social, and demographic spaces of the state.

Therefore, scientists identify risks with danger, that is, the consequences in this case can only be negative. Schmid and Schömann (2004), highlighting the educational determinants of migration risks, note that intellectual migration as the latest characteristic of socio-demographic processes (SDP) combines two opposite processes: the growth of human capital of Ukrainian citizens and the washout of Ukrainian SDPs since the results of their labour activities are used by other countries.

In a Western European scientific school, the study of social risks has become especially popular in connection with the study of the so-called "new risks". Their appearance was primarily determined by factors of de-industrialization of the economy, an increase in the share of services in total employment, instability in labour markets, increased instability of family structures, as well as the privatization processes of welfare states (Whelan and Maître 2008).

"New social risks" were defined as "risks faced by people throughout life as a result of economic and social changes associated with the transition to a post-industrial society" (Zhang (2011)). In the framework of this approach, social risks were identified as follows: universal risks; group/class risks; risks of individual stages of life; risks of intergenerational relationships.

There is also no less common approach according to which new social risks were classified as follows (Taylor-Gooby 2004b):

- common significant risks (people who are not able to work due to illness or old age)
- group specific risks I (people with insufficient skill level due to the onset of structural changes);
- group specific risks II (people who have an indefinite return on obtaining higher education);
- group specific risks III (people who may lose income as a result of having a baby).

Liu et al. (2016), not limited to the conceptual vision of social risk, propose a group model matrix with the help of which he explains people's behavioural strategies (by stereotypes of behaviour—advocates of equality, individualists, hierarchists, fatalists, and hermits). The basis of human behaviour is the cultural traditions of society and education. They also determine the probabilistic characteristics of social risks.

Ferragina et al. (2015) believe that the interconnection of sociocultural structure, awareness of risks, and their management regime should be the central element of risk-logical research. By social structure, the researcher proposes to understand the common boundaries or matrices in which typical patterns of solidarity, fragmentation, and conflict are formed. In addition, the "modern", "hyperreflexive", and "non-traditional" social systems are distinguished.

The study of social risk from the point of view of its perception by respondents today is carried out mainly within the framework of the psychometric paradigm, which clearly distinguishes between the concepts of danger and risk. The first ones are meant to understand the threats that people evaluate at a time when risks are a quantitative expression of the consequences of the danger and can be expressed as the conditional probabilities of the damage. The study of risk can also be carried out by constructing theoretical models, the structure of which are based on subjective judgments and assumptions. Representatives of different social groups should act as respondents; they are proposed to rank three dozen technologies in accordance with the growing danger. Although the results will vary significantly among themselves, however, they correlate well with statistics on mortality that accompanies a particular technology. This allows for the conclusion that there is a direct relationship between expert risk assessments and the number of deaths at work.

A review of the scientific discourse revealed that some important problems, including the study of the social aspects of the risks arising in the process of labour migration and cross-cultural interaction, remain without the attention of scientists and practitioners. For instance, there is a lack of conceptual ordering, there is no comprehensive classification of risks arising from cross-cultural interaction of migrants with nationals of host countries, and there are few verified tools for measuring migration risks in the context of migrant employment within the organizational and business culture of host societies.

To determine the scope of the study, the definitions of the main definitions are given.

In the understanding of the UN Convention, a labour migrant is a person who will engage in, engage in, or have engaged in paid activities in a state of which he or she is not a citizen (OHCHR, 1990).

Experienced labour migrants are labour migrants who have been working outside the donor country for more than five years.

The main working hypothesis of the study is based on the assumption that an insufficient level of sociocultural competence contributes to the formation of imaginary obstacles that do not have objective premises. In addition, we assume that avoidance of uncertainty and risks and delegation of authority to intermediaries deprives labour migrants from Ukraine of initiative, self-confidence, the ability to make independent decisions, seek and use feedback from the environment. All this becomes the basis for neglecting the formation of sociocultural competence in intercultural interactions and the formation of new migration risks and social threats.

The purpose of the study is to find out the differences in the perception of obstacles and risks that arise in the process of work abroad among experienced and potential labour migrants from Ukraine within the cognitive, behavioural, and emotional components of their intercultural competence.

## 2. Materials and Methods

A number of general scientific and special sociological methods are applied in the work, namely logical-semantic-for analysis and deepening of the conceptual framework of the external migration notion, comparative analysis of the results of statistical and specific sociological studies of migration, modeling risks faced by labour migrantsstructural and functional analysis in identifying social risks faced by labour migrants in a foreign cultural environment, as well as a number of statistical methods in the analysis of results of the author's empirical research.

The methodology for assessing social risks must comply with the European system of social indicators, which involves: measuring social processes in the coordinates of identity, unevenness, consolidation, and conflict; justification and combination of new dimensions or indicators; use of the most appropriate databases and results of national studies (Esteves et al. 2017).

Based on the above provisions, the task of assessing social risks will be reduced to assigning a specific numerical value to each riskogenic factor, which will determine the weight of each factor in their totality. To do this, for each questionnaire, it is necessary to construct a Saati pairwise comparison matrix and find in it the maximum eigenvalue and its corresponding normalized vector. Elements of this vector correspond to the weight of each factor and reflect the responses of each respondent. The last step will be to find the average value of all vectors, the resulting vector will contain the weight of latent factors according to the survey (Chen and Khumpaisal 2009):

$$G = \omega_1 \times F_1 + \omega_2 \times F_2 + \ldots + \omega_r \times F_r \tag{1}$$

where $G$ is the resulting vector of latent risk factors; $\omega$—weight and $i$—that weight factor; $F$—detected latent factors

The approach of World Bank experts should be noted. It is based on the fact that risk and its dimensions are traditionally associated with variability in income or consumption. Typically, such measurements are based on variation or mean-square deviation. However, if it is necessary to study the impact of risks on the well-being of individual segments of the population (in particular, the poor), then this approach in most cases becomes unacceptable. In order to adapt this methodological approach, it is proposed to derive three indicators of social risk, based on only those goals that households pursue in the risk management process. If the goal is (Vyacheslav L. Baburin et al. 2014):

- minimization of the sizes of possible welfare losses, then such a solution does not need information about the probability, just a set of loss functions is enough; the magnitude of the risk will represent quantitative indicators of losses—min, max;
- minimization of the likelihood of a drop in consumption below a certain threshold level. To make a decision, information on the expected income from alternative activities and the threshold value of consumption is needed; the magnitude of the risk will represent the probability—min Pr $(c_t \leq c_{min})$;
- maximization of expected return with a certain level of its variability. To make a decision, information on the advantages in relation to risk, expected return, and distribution of various types of assets is necessary; the risk value will be displayed by the mean-square deviation—max V $(\mu; \delta)$.

Also, one of the key problems highlighted by World Bank experts is the fact that risk assessment should be carried out according to the ex-ante approach (Marius R. Busemeyer 2017), that is, in terms of temporality (short, medium, or long term). However, most of the estimated indicators given by them are based on the ex-post approach, that is, on data obtained in past periods excluding forecast

estimates of future events and the conditions in which they will take place, which creates additional difficulties in implementing a risk management optimization policy.

To analyse and assess social risks of a high or catastrophic level, in addition to the risk and benefit analysis method. In our opinion, it is advisable to use methods based on the foundations of probability theory and mathematical statistics based on its postulates. Since this type of risk is mainly destructive, these risks are difficult to quantify and their characteristics are probabilistic. This is because the qualitative parameters of possible losses can be so significant that the society will simply not be able to create appropriate reserve funds to eliminate losses from an adverse event.

The assessment of social risks in the social plane proceeds from the fact that their specific factors in this aspect are risk perception, relativity of risk, and risk justification. In the process of analysis, one should take into account the fact that the boundaries of social risk are perceived depending on the maximum and minimum acceptable levels of risk for various population groups, different territories, and the like. Therefore, it is necessary to ensure the coherence and relevance of risk assessments and determine its impact on the most vulnerable groups of the population.

Certain aspects of social risk can be assessed by independent experts, for which different risk assessment methods or even different economic and social criteria for the impact of risk can be used, which can lead to disagreements in assessing the overall risk. This determines the need to ensure the same (comparable) standards, criteria, and assessment methods. Social assessments of a certain type of risk should not be provided without comparison with assessments of other risks since each risk in society operates simultaneously with other "competing" risks. Therefore, efforts to assess and reduce a certain type of risk should be relevant to its level relative to other risks. The expected impact of various risk reduction options must be assessed by the degree of its impact on the most vulnerable groups of the population.

The inclusion of risk equity assessments in risk management is becoming an increasingly important issue. Uneven income distribution and differentiation of the living standards of the population should be taken into account when assessing risks, especially when certain types of risks are higher for vulnerable segments of the population. It should be borne in mind that justice in the distribution of social risks is often the subject of attention of various political forces.

The issue of assessing risks and the degree of vulnerability of certain segments of the population today is quite controversial. This is due to the need to obtain information about who is most at risk and how unfavorable the outlook seems. Such estimates are necessary to calculate the costs associated with risk, distributional consequences, and ensuring targeted measures aimed at minimizing them.

Moving from universal approaches to the analysis and assessment methods that will directly concern social risks in the labour sphere, it should be noted that they are very limited and have their own characteristics.

The empirical basis of the work consists of the materials of the author's research, carried out using an online survey (n = 841) of potential and experienced migrant workers. The study was conducted from 25 January 2019 to 24 February 2019.

A questionnaire was developed for the empirical phase of the study. The questions' format in the first part of the questionnaire was common to all participants in the program and was designed to assess the cross-cultural competency of migrants and potential migrants working or preparing to work abroad. The joint part of the questionnaire also contained a 5-point scale, where "5" points mean "very important" and "1" not at all important. Questions of the first part of the questionnaire were divided into blocks, which were designed to determine the frequency of communication with foreigners and geography of cross-cultural contacts of respondents, determine the channels of exchange of information with foreign partners. The second part of the questionnaire dealt with issues that were important for each participating country.

The information was gathered through a combination of a postal survey using the emails of experts in the field of international labour migration and an online survey on social networks of the Internet. The poll was conducted from 25 January 2019 to 24 February 2019. A SurveyMonkey

sociological research platform was created for the survey. With the help of SurveyMonkey system, an e-mail link was sent to 2500 e-mail addresses. Since the return of emails (respondents' participation in the first phase of the survey) was 19.1%, the questionnaire links were posted on the social groups of the Facebook and Google+. This geographical location was determined by the following logic—the indicated regions represent a certain gradient in the share of migrant workers.

The survey was controlled by a double-sample method based on respondents' gender and age. The survey was discontinued when gender and age in the target group stabilized at a level consistent with that of the general population.

The Ukrainian part of the questionnaire was aimed at studying the obstacles facing international labour migrants, assessing the most important knowledge about the country's historical and social phenomena, determining means of communication with foreign partners, measuring the frequency of appeals to foreign sources for resolving work situations (Appendices A–C)

To assess the extent of internal consistency of issues, the Cronbach's alpha coefficient was used, which was calculated in the PSPP software package. For most of the answers to the questionnaire, the Cronbach's alpha coefficient was not lower than 0.85, which indicates the internal consistency of the indicators used (Appendix C)

The odd thing, you know, is that 841 respondents took part in the survey. Among them, 347 respondents, or 41.26% of the total number of respondents, belonged to the target group of the study, i.e., they used to go abroad to work. The control group included 347 respondents who had never traveled abroad to work. During the survey, 147 respondents, or 17.48% of the total number of respondents, did not answer the question about the frequency of going abroad to work. They were not eliminated from the array of respondents because they included some respondents who acknowledged their work abroad through additional questions and determining their real location by IP address. A group of respondents who shied away from answering the question of frequency of going abroad could be useful for a more complete analysis of the risks of migrant workers. The statistical validity of the comparison of the respondents of the target and control groups was performed using the Student's *t*-test for comparison of the averages. The differences were statistically significant at the level $p = 0.01$.

Total sampling consists of 53.31% of female respondents and 46.69% of male respondents, which corresponds to the distribution by gender of the general population. The gender distribution in the control group of respondents is 63.82% of female respondents and 37.18% of male respondents, which reflects the general tendency of more active participation of women in the survey. The gender distribution in the group that did not answer the question of frequency of going abroad to work is 56.46% of female respondents and 43.54% of male respondents. It should be noted that there is no questionnaire where the gender of the respondent would not be indicated (Table 1). This may indicate a high level of respondents' interest in answering the questionnaire.

**Table 1.** Distribution of respondents by gender (N, %).

| Gender | Target Group | Control Group | Not Answered the Questions | Total |
|--------|:---:|:---:|:---:|:---:|
| Females | 185 | 218 | 83 | 486 |
| | 53.31% | 63.82% | 56.46% | 57.79% |
| Males | 162 | 129 | 64 | 355 |
| | 46.69% | 37.17% | 43.54% | 42.21% |
| Total | 347 | 347 | 147 | 841 |
| | 100.0% | 100.0% | 100.0% | 100.0% |

Source: the author's research data.

The distribution of respondents by age in the target group as a whole corresponds to the demographic structure of the general population. A small shift in the number of young people under

25 in the control group of respondents cannot significantly affect the result of its comparison with the target group. Respondents of all other age groups are represented in the data set in proportion. The number of questionnaires skipped when answering the age question does not exceed 1% of the total number of respondents (Table 2).

**Table 2.** Cross tabulation of respondents' age and belonging to target and control groups (N, %).

| Age | Target Group | Control Group | Not Answered | Total |
|---|---|---|---|---|
| Below 25 years old | 103 | 155 | 69 | 327 |
| | 29.68% | 44.67% | 49.29% | 39.21% |
| From 25 to 35 years old | 91 | 76 | 30 | 197 |
| | 26.22% | 21.90% | 21.43% | 23.62% |
| From 35 to 50 years old | 94 | 71 | 22 | 187 |
| | 27.09% | 20.46% | 15.71% | 22.42% |
| Above 50 years old | 59 | 45 | 19 | 123 |
| | 17.01% | 12.97% | 13.57% | 14.75% |
| Total | 347 | 347 | 140 | 834 |
| | 100.0% | 100.0% | 100.0% | 100.0% |

Source: the author's research data.

The most difficult part of the survey was the procedure for identifying respondents at the place of work. 83 respondents indicated their place of work abroad, or 23.92% of the target group and 9.99% of the total number of respondents. Geography of labour migration of the respondents includes 22 countries: 12 people (14.46%) work in Russia, 11 people (13.25%) in the Czech Republic, 7 people (8.43%) in Poland, 6 people (7, 23%) in Slovakia, while 5 people work in Austria, Germany, and the USA. In addition, migrants from Hungary, Spain, Italy, France, Belgium, Belarus, Vietnam, the United Kingdom, India, Canada, the Netherlands, Romania, Turkey, Montenegro, and Switzerland are among the respondents. The number of respondents from the target group who work abroad was only 15.85%. All other respondents working abroad belong to two groups: (1) those who did not answer the question about the frequency of travelling abroad for work (7.48%); (2) respondents from the control group (4.90%), who claim that they do not go to work abroad at all.

## 3. Results

New social risks for labour migrants emerge outside the conflict between labour and capital, which has been characteristic of industrial society. The nature of new social risks is manifested by the conflict between incompetence and professionalism of employees.

Social risks are formed, as mentioned earlier, under the influence of internal and external factors. In our study, we focus on the social risks associated with the internal obstacles of labour migrants. Namely, the research structure is related to socio-cultural competence. Socio-cultural competence consists of cognitive, emotional, and volitional (behavioural) elements and, in the context of international migration, is manifested primarily in the intercultural interaction of migrants and residents of host countries. The cognitive aspect of intercultural competence includes the knowledge that a person receives most often through targeted training. This fundamentally distinguishes the cognitive aspect of competence from the behavioural (volitional) aspect. If the result of targeted learning is attributed to the cognitive aspect of intercultural competence, then experience is always an indicator of the behavioural aspect of competence. The result of experience is skills. The third component of sociocultural competence is emotional. Its main indicator is the relation to other identities, cultural values. Most often, this component is manifested as (in) tolerance to culture and race differences.

Social risks of international labour migrants are shaped by their own choices and may lead to loss of a significant part of their competence or initiative components. First of all, it concerns self-confidence, ability to make independent decisions, seek and use feedback from the environment. Migrants' work abroad in the third segment of the labour market is fraught with the loss of a life strategy aimed at social success.

International migrant workers should be aware that by their own choice they can create a danger not only to their immediate circle, their own families, but also to the entire socio-economic potential of the country of origin. The choice of labour migration as a strategy may be insufficiently motivated due to an underestimation of the level of willingness to migrate and uncertainty while in a different cultural environment.

In contrast to objective obstacles, imaginary ones appear at all three stages of labour migration (Yu et al. 2017): (1) the stage of psychological readiness for movement; (2) the stage of the spatial displacement itself; (3) the stage of adaptation to new conditions in the host countries. Imaginary obstacles can manifest themselves in the cognitive, emotional, and behavioural aspects of the activities of labour migrants, highlight imaginary risks and social threats.

Correlation regression analysis revealed a strong direct link between the short-term stay of labour migrants from Ukraine abroad (1–3 months) and the uncertainty avoidance index (R = 0.671). Between the long-term stay of labour migrants from Ukraine abroad (3–6 months) and the uncertainty avoidance index, there is a weak relationship with the correlation coefficient (R = 0.179).

Between the long-term stay of labour migrants from Ukraine abroad (6–12 months) and the uncertainty avoidance index, there is a strong feedback with a correlation coefficient R = −0.604. Figure 1 shows a steady regression over the residence time, from a strong direct correlation to a strong inverse correlation.

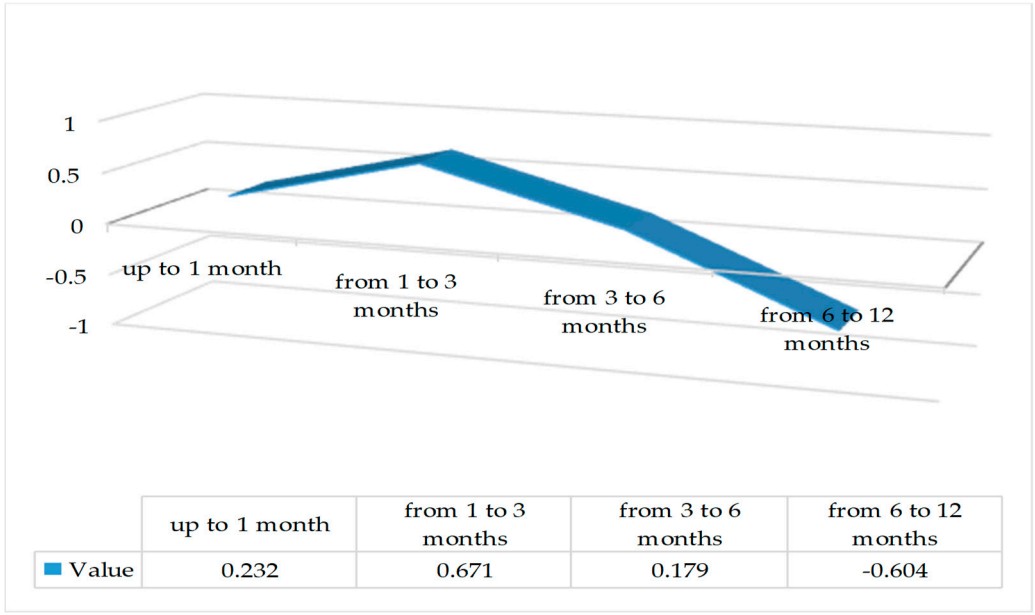

|  | up to 1 month | from 1 to 3 months | from 3 to 6 months | from 6 to 12 months |
|---|---|---|---|---|
| ■ Value | 0.232 | 0.671 | 0.179 | -0.604 |

**Figure 1.** Diagram of the correlation between the term of stay and the uncertainty avoidance index of the labour migrants from Ukraine. Source: developed by the author based on (Migration Profile of Ukraine 2011–2015).

The correlation coefficient between the stay of labour migrants abroad for up to one month and the uncertainty avoidance index somewhat falls out of the general trend. It is lower because short-term stay is more common among men. The general trend can be defined as a reduction in the length of stay of labour migrants from Ukraine abroad, depending on the attitude to risks and the situation of uncertainty avoidance. The more restrained the attitude of labour migrants towards risks, the less time they spend in the host country.

The results of the correlation analysis showed the opposite trend in the relationship between the length of stay of labour migrants abroad and the long-term orientation index (LTO). Figure 2 shows the correlation of two opposite trends, which confirms the inverse relationship between the avoidance of uncertainty index and the long-term orientation of labour migrants from Ukraine.

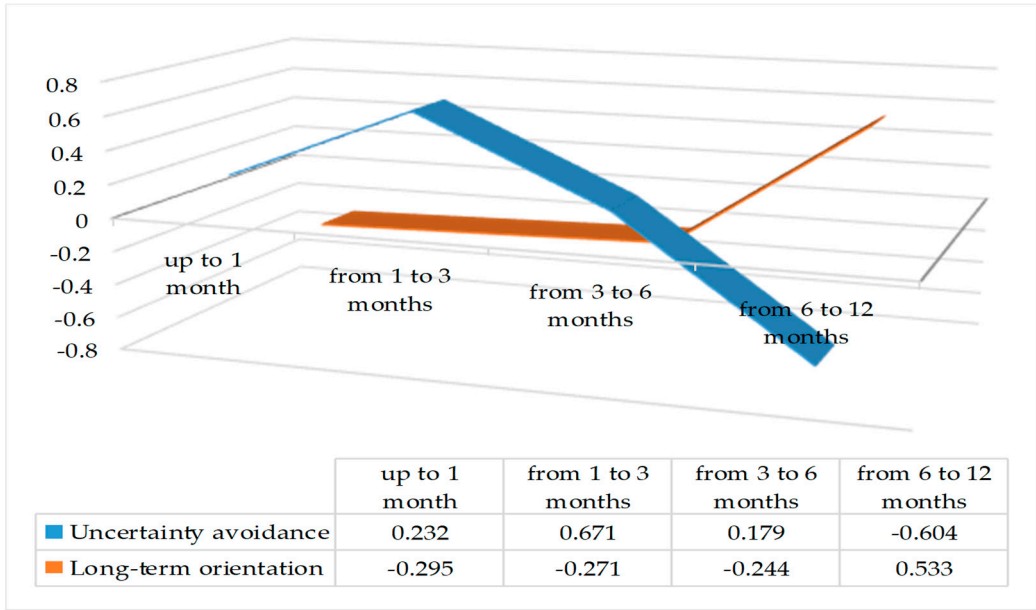

| | up to 1 month | from 1 to 3 months | from 3 to 6 months | from 6 to 12 months |
|---|---|---|---|---|
| Uncertainty avoidance | 0.232 | 0.671 | 0.179 | -0.604 |
| Long-term orientation | -0.295 | -0.271 | -0.244 | 0.533 |

**Figure 2.** Diagram of correlation relationship between the length of stay, the uncertainty avoidance index, and the long-term orientation of labour migrants from Ukraine. Source: developed by the author based on (Migration Profile of Ukraine 2011–2015).

The long-term orientation of labour migrants is associated with a low level of uncertainty avoidance and terms of work abroad from 6 to 12 months. Thus, the resettlement of labour migrants for permanent residence (emigration) depends not only on the duration of their work in the recipient country but also on their long-term orientation and willingness to take risks. The volume of these migrants does not exceed 14% of the total. A strong rejection of uncertainty in the organizational culture of migrants from Ukraine holds them back from the risks of relocation to other countries.

Attempts by migrants from Ukraine to avoid risks or take on only carefully calculated risks are complemented by a strong direct correlation between the uncertainty avoidance index and the index of distance from governance (R = 0.695). The direct correlation between the indices means an a priori recognition of social inequality and injustice in relations. This makes status important for migrants. During negotiations and the conclusion of cooperation agreements, the hierarchy and a clear position on authority or subjugation between the parties to cooperation will be of great importance for migrants from Ukraine. The distinction between organizational cultures is complemented by a high level of collectivism of the organizational culture of Ukrainian labour migrants, which means trust exclusively in the loyal environment.

The identification of possible obstacles allows international labour migrants from Ukraine to envisage prospects for communication with representatives of cultures of the host country. Unlike risks and threats, obstacles in intercultural communication should be considered as the most effective indicator of research of the entire security system. The identification of obstacles is maximally accessible for all groups of respondents and causes minimal differences in interpretation and ambiguity of interpretations.

Our research is based on the assumption that an insufficient level of cross-cultural competence facilitates to the attribution of imaginary impediments and creates grounds for avoiding uncertainty and risk. The hypothesis was tested within the framework of the research program on cross-cultural

competence of migrant workers, which was implemented by an international research team with the participation of the author of this work.

A question block on barriers to communication with the host country residents was contained a question to assess the importance of the language barrier. Responses of labour migrants who indicated and not indicated the host country were found similar. In particular, similarity shows up in a refusal to recognize poor command of the host country language. None of the respondents in the group of experienced labour migrants admitted that lack of language skills was an obstacle to communicating with foreign employers. Insufficient knowledge of the host country language was recognized only by those respondents who had little or no experience of working abroad. Positions of experienced and potential labour migrants and those who have neither experience abroad nor such intentions also differ significantly in their perceptions of other obstacles in communicating with foreign employers.

The procedure for identifying respondents at their place of work has been most difficult aspect of the study. In total, 83 respondents indicated the place of their work abroad, or 23.92% of the number of respondents in the target group, and 9.99% of the total number of respondents. The geography of labor migrations of the surveyed respondents includes 22 countries: 12 people (14.46%) work in Russia, 11 people (13.25%) in the Czech Republic, seven people (8.43%) in Poland, six people (7. 23%) in Slovakia, while five people work each in Austria, Germany, and the USA. In addition, among the respondents are labor migrants from Hungary, Spain, Italy, France, Belgium, Belarus, Vietnam, Great Britain, India, Canada, the Netherlands, Romania, Turkey, Montenegro, and Switzerland. The number of respondents in the target group whose place of work is located abroad is only 15.85%. All other respondents working abroad belong to two groups: (1) those who did not answer the question about the frequency of travel to work abroad (7.48%); (2) respondents from the control group (4.90%) who state that they do not go to work abroad at all.

In particular, those who have never worked abroad recognize the lack of communication skills as a major barrier to communication with foreign employers, and for the experienced labour migrants a lack of communication skills does not appear to be such a significant obstacle (Table 3).

**Table 3.** The main obstacles in communication with foreign employers (N, %).

| What is an Obstacle in Communicating with Foreign Employers? | Did not Work Abroad | Potential Labour Migrants | Experienced Labour Migrants |
|---|---|---|---|
| Insufficient awareness | 77 22.19% | 117 33.72% | 19 22.89% |
| Lack of communication skills | 125 36.02% | 114 32.85% | 18 21.69% |
| Disrespect for cultural values | 5 1.44% | 11 3.17% | 4 8.82% |
| Ignorance of the language | 12 3.45% | 7 2.02% | 0 0.00% |
| Other obstacles | 65 18.73% | 48 13.83% | 14 16.87% |
| Several of the mentioned obstacles | 25 7.20% | 26 7.49% | 7 8.43% |
| There are no obstacles | 20 5.76% | 15 4.32% | 8 9.64% |
| No answer | 18 5.19% | 9 2.59% | 13 15.66% |
| Total | 347 100.0% | 347 100.0% | 83 100.0% |

Source: the author's research data.

Responses to the question about the ability to use a foreign language in the course of professional activity showed that a higher estimation of their ability to speak a foreign language in the course of professional activity is characteristic of respondents with less work experience abroad. Almost a third

of respondents from the group of potential migrant workers fully agree on their ability to communicate in a foreign language. The level of complete confidence in their own ability to communicate in a foreign language is slightly lower (26.51%), but it is almost twice as high as the indicator of confidence in the ability to communicate in a foreign language than those respondents who never worked abroad.

High appreciation of potential migrants' ability to use a foreign language in the professional activity is associated with a low appreciation of the importance of knowing a foreign language in the work environment. This is an imaginary obstacle for migrant workers.

Imaginary (intermediate) obstacles of labour migrants are a concept introduced into the sociology of migration by Mangalam and Schwarzweller (1970) in order to improve the two-factor model of migration and reduce the analytical rigidity of the "attraction-repulsion" model.

Imaginary, or intermediate, obstacles of labour migrants are a phenomenon of individual and mass consciousness that influences migration intentions and migration behaviour.

In particular, the estimation of the importance of knowing a foreign language in the work environment by experienced migrants is much lower than that of the respondents who had never worked abroad (Figure 3).

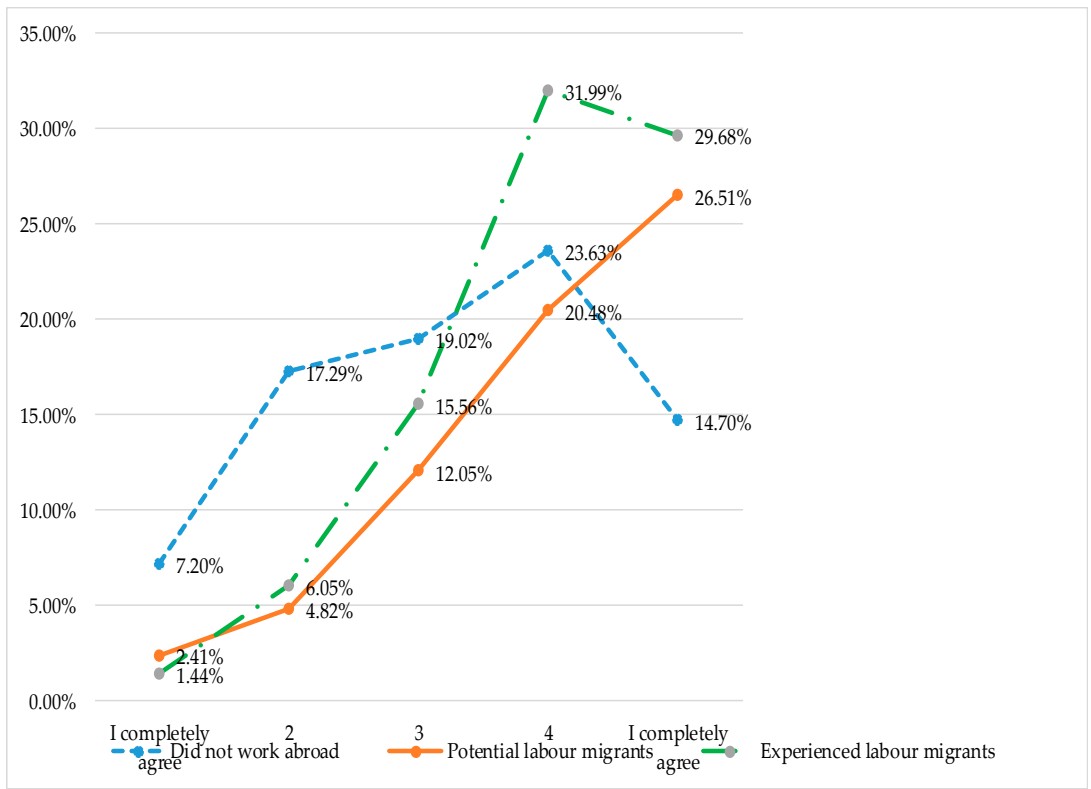

**Figure 3.** Distribution of responses of experienced and potential labour migrants regarding the ability to use a foreign language in their professional activity, % (author's research data).

Undoubtedly, knowing the language of communication with foreign employers and understanding the importance of using the language in the work environment are different aspects of studying the language barrier of labour migrants. The issue that is able to integrate all identified aspects of the problem is the degree to which respondents agree that poor language skills is an obstacle that most often arises when communicating with foreigners. This idea is supported by about a quarter of respondents who have little or no experience of working abroad. The same quantitative indicators on the language barriers of labour migrants were obtained in the study using public opinion polls. Experienced migrant workers perceive the language problem much less acute.

Less than half of experienced migrant workers believe that knowledge of the host country's language is very important for successful interaction with the host country residents (46.99%). At the same time, potential labour migrants have a significantly higher assessment of the importance of knowing a foreign language (64.55%). This may be indicative of the superficiality of the views of less experienced migrants, mentioned by the researchers of the "Rating" sociological group. Abstract perceptions of working conditions abroad promote increased intentions for labour migration, but do not reduce the risks and vulnerabilities in the unfamiliar circumstances of working abroad

Knowledge of a foreign language is considered important for successful communication in the work environment by 46.99% of respondents from the group of experienced migrants and 64.55% of those with less experience of migration abroad. In terms of language experience, 15.66% of experienced migrants and 27.38% of those who have never worked abroad reported language problems.

Somewhat paradoxical results are obtained in the group of experienced migrants: only 26.51% of the respondents from this group believe that they are able to use a foreign language in their professional activity, but no respondent from this group indicated that lack of language skills is an obstacle to communication with foreign employers.

The situation with the notion of importance of managing new situations in the process of cross-cultural interaction is peculiar. Significant importance of managing new situations in the process of cross-cultural interaction was reported by only 15.66% of experienced migrant workers and 23.63% of respondents who had never worked abroad. Differences between the other elements of cross-cultural competence of the two groups reach 27–29%. In particular, high differences are recorded in the assessment of awareness of important historical and social phenomena of host countries, tolerance for culture and race differences, respect for values of other cultures. Significant differences in estimates of the importance of different elements of cross-cultural competence indicate the complex latent structure of this phenomenon.

The special attention of labour migrants in the language barrier is formed in Ukraine and not beyond. A vision of the severity of this problem is an integral part of the problems and risks of labour migrants. Most often, risks are imaginary, do not exist in reality, and are not essential for adaptation abroad. A false, too generalized, stereotyped perception of working abroad leads to the formation of obstacles in the form of imaginary risks. One of the perceptions about risks in the process of external migration is the perception of a language barrier. In turn, the prevalence of such perceptions, especially among people without work experience abroad, contributes to the active use of intermediaries. Moreover, almost the entire market of such intermediary services was recently controlled by intermediaries who acted "in the shade" outside the legal field, which, accordingly, entailed a risk of violation of labour rights of migrants. In the event of illegal employment, the imaginary obstacles of potential labour migrants turn into real risks of labour migration, and numerous labour migrants become victims of fraudsters.

A stable social stereotype is the opposition of motivation to foreign migration to the motives of patriotism. The position of opposing patriotism and international labour migration finds arguments in recognizing the inevitable transformation of the ethnocultural identity of labour migrants.

Defining communication skills deficiency as impediments to the interaction of labour migrants with foreign partners leads to the conclusion of there being a certain advantage of the behavioural aspect of migrants' cross-cultural competence over the cognitive aspect. Communication skills are a reliable indicator of the behavioural aspect of cross-cultural competence. They are able to increase migrants' chances of gaining cross-cultural competence.

## 4. Discussion

Labour migrants assess risk by calculating the balance of profit and loss, positive and negative effects of migration for the immediate environment and family. Measuring dangers to the socio-economic perspectives of the originating society at the institutional level is inconsistent with individual estimates of labour migrants. The main obstacle to distinguishing between social risks and

dangers is their traditional understanding aimed at compensating for losses, and identifying them with socially significant objectively recognized social circumstances as recognized by society.

The social risks faced by labour migrants are shaped beyond the conflict between labour and capital, which has been characteristic of industrial society. The nature of new social risks is manifested in the conflict between (in)competence and (non)professionalism of employees. New social risks are shaped by the individual choices of international labour migrants, have manifestations at both individual, group, and social levels, and can be defined as the likelihood of negative social consequences as a result of migration behavior. International labour migrants can pose a danger by their choice not only to the immediate environment, their own family, but also to the entire socio-economic potential of the country of origin. The choice and risks of international migrants may be a conscious choice or insufficiently motivated due to an underestimation of the level of willingness to migrate and uncertainty about being in a different cultural environment. One of the aspects of migration risks and communication competencies is the socio-cultural aspect, which is the focus of this research.

The study of labour migration abroad must certainly be based on a clear differentiation of migration sentiments, migration intentions and, in fact, migration behavior. Consideration should also be given to the complexity of each of these phenomena, which may manifest themselves at the intellectual, emotional, or volitional levels. Due to the decisive importance of the emotional and volitional components of the decision, growth of migration sentiments of economically active population significantly outstrips the real readiness for international labour migration.

The feeling by labour migrants from Ukraine of their inferiority ("second-rate") in the EU countries is supported by the existing "institutional framework", namely the licensing of certain types of professional activity, residence rules, etc. (Koshulko 2015).

Such an attitude hinders the integration process and secures the status of outsiders for migrants. In France and Germany, there is a three-tier system: full political rights belong to citizens, truncated political rights belong to citizens of other countries of the European Union, and as to the third world representatives (not in the EU), they don't have any political rights at all (Standing and Charter 2015).

However, at the same time, they also suffer from the uncertainty of their position in the labour market, i.e., disqualification, a sense of inferiority for which they are trying to compensate by the disorganization of social relations or anomie.

It is proved that the risks of ethnocultural marginalization of labour migrants from Ukraine are exaggerated. A comparative measurement of the importance of the values of Ukrainian society for migrants with different periods of stay abroad showed a steady interest in these values in all groups of migrants. The importance of Ukrainian cultural values does not only not decrease with the length of stay abroad, but also grows (Fedyuk 2009).

In our opinion, in order to solve the problems revealed in the study, it is necessary to change the professional training of specialists in Ukraine. Thus, more attention should be paid to the study of foreign languages, changing educational standards of practical training in special disciplines, and the introduction of psychological cases in order to overcome psychological barriers in communication.

With the introduction of quarantine in Ukraine in 2020 due to the epidemiological situation of COVID-19, the government did not want to let albor migrants go back to European countries. However, instead of this, it offered low salaries for vacancies. So, in the field of construction in Ukraine, the salary range is 200–500 dollars. In the EU member states, they offer 1000–2000 dollars for labor migrants. Of course, if the government wants to leave its able-bodied citizens in the country, it is necessary to radically change the socio-political situation, i.e., to reduce taxes for businesses and citizens. If in their own country citizens do not have decent wages, or enjoy the protection of civil rights, then they will feel "inferior" in other countries.

A significant social risk for labour migrants from Ukraine is the loss of initiative and motivation for intercultural dialogue with residents of the host country. These risks become the basis for interethnic friction, subject labour migrants to secondary roles in the workplace, and force them to accept unfavorable "conditions of the game" on the part of intermediaries and residents of the host country.

This situation deepens the risk of the precarization of labour migrants and causes various forms of disorganization of social relations.

It has been found that, under the conditions of a post-industrial society, due to the expansion of choice and an increase in the level of uncertainty, the significance of diverse social risks increases. The social risks faced by labor migrants are formed outside the conflict between labor and capital, which was characteristic of an industrial society. The nature of new social risks is manifested due to the conflict between the (in)competence and (un)professionalism of employees. It has been revealed that the main aspect of the social risks faced by Ukrainian labor migrants abroad is the discrepancy between the Ukrainian organizational cultures and that of the host countries. Based on the data of empirical sociological research, it has been revealed that these inconsistencies, in particular, relate to the attitude towards uncertainty and risks.

## 5. Conclusions

It has been established that, in the post-industrial society, due to an increasing versatility of choice options and growing level of uncertainty, the importance of diverse social risks is increasing. One of the challenges of the post-industrial society is a radical change in the temporal structures of society, which is an indicator of risks. They make it possible to see the shifts between the "expected future" and the "true future", to mark future threats in the present. The social risks of international labour migrants whose strategic choice is directed at self-fulfilment are linked to an increased chance of achieving social success. Distinguishing between risk and danger can improve migrants' competence, prevent wrong decisions, and restrain government policy from excessive regulation. Risk identification based on a priori capabilities, statistical calculations, or intuitive estimates is based on the daily experience of migrant workers.

It is found that the key points of the comparative analysis of social risks faced by migrant workers open up the prospects for improving the methodology for researching social (and socio-cultural in particular) risks. First of all, it concerns development of research methods for migration decision-making, mechanisms of uncertainty prevention and participation in cross-cultural communication.

Further elabouration requires definition of "the fictitious obstacles" concept that is outside the cross-cultural competence of migrants. The study results proved the prospect of exploring the (im)balance between cognitive, emotional, and behavioural components of the cross-cultural competence of migrant workers in order to prevent new social risks beyond the competence of the state.

The author's research is distinguished by several features. A new vision of the socio-cultural risks of labor migrants is proposed in this work for the first time, which is not identified with the normative legal fixation of socially significant circumstances, but reflects the informational nature of new social risks that become relevant in the transition to a post-industrial society. It has been proved that a high coefficient of uncertainty avoidance and risk prevention of labor migrants from Ukraine increases their vulnerability, subjects them to the secondary roles in production, and forces them to accept unfavourable working conditions on the part of intermediaries and employers of the host country. Predictably, the likelihood of an increase in the risks of losing "motivated abilities" is based on the imbalance between the cognitive, emotional, and volitional components of the intercultural competence of labor migrants from Ukraine and the strengthening of their sense of inferiority.

It has been established that most migrants recognize the insufficiency of their knowledge and skills necessary for communication with foreigners, but they believe that enhancing their cross-cultural competence would not contribute to their success abroad. By means of comparative research methods of several groups of actual and potential migrant workers, it was proven that obstacles and risks of labour migration differ, depending on the country of destination of migrant workers. In communication with residents of Eastern European countries, the major obstacle is considered to be a lack of communication skills, whereas a lack of awareness is considered to be the major obstacle in communication with residents of Western European countries. Taking into account the feedback of residents of different countries, it is proven that a lack of communication skills creates a greater barrier to communication.

**Author Contributions:** Conceptualization, A.K. and A.L.; Data curation, H.O.; Formal analysis, I.D. and H.O.; Funding acquisition, A.L.; Investigation, L.H.; Methodology, A.K.; Project administration, L.H.; Resources, L.H.; Software, A.K.; Supervision, A.K.; Validation, A.L. and I.D.; Visualization, A.L.; Writing—original draft, I.D.; Writing—review and editing, H.O. All authors have read and agreed to the published version of the manuscript.

**Funding:** This research received no external funding.

**Conflicts of Interest:** The authors declare no conflict of interest.

## Appendix A

Structure and content of the questionnaire of the sociological study "Socio-cultural competence of potential labour migrants and migrants who have abroad work experience in the context of migration risks"

The questionnaire consisted of 10 thematic blocks, which were operationalized in 88 questions. After checking the reliability of the Cronbach's alpha coefficient, there were 83 questions left.

**Table A1.** Blocks of questions.

| Block of Questions | The Importance of Questions and Their Place in the Operationalization of the Concept of "Sociocultural Competence of Labour Migrants in Intercultural Interaction" |
|---|---|
| 1. Questions designed to determine the frequency of communication with foreign citizens and the geography of intercultural contacts (4 questions) | The block of questions is aimed at identifying workers who need to develop intercultural competence and establish the geography of intercultural contacts of migrants. It contains questions about the frequency and geography of contacts with foreign citizens: at work; when they are abroad in business; when traveling abroad countries visited by respondents. |
| 2. Questions on respondents' perception of the need to develop their socio-cultural competence in interactions in the work environment (4 questions) | The block of questions concentrates on how respondents perceive the development prospects of their organization and their contribution to this development: <br><br> perception of respondents about the impact of their socioculturalcompetence on a career (individual level) perception of respondents about the impact of their sociocultural competence on the success of organizational activities (organizational level) <br><br> assessment of the development of international contacts of the organization (organizational level) understanding of the development of sociocultural competence of workers (individual and organizational level). |
| 3. Questions designed to assess the cognitive dimension of socio-cultural competence of respondents (30 questions) | The block refers to the knowledge of respondents that are indispensable for demonstrating sociocultural competence and relate to the awareness of the characteristics of social, historical phenomena of their own and foreign culture, understanding the processes of interaction between an individual and society (organizations), causes of cultural misunderstandings, etiquette requirements, taboo themes, common stereotypes and etc. |
| 4. Questions aimed at assessing the socio-economic status of respondents (5 questions) | This block of statements is aimed at assessing the skills that are important for the sociocultural competence that a person demonstrates in communicating with foreign citizens: the ability to put their knowledge into practice; establish and maintain intercultural contact; distinguish cultural differences and respond flexibly to them. |

**Table A1.** *Cont.*

| Block of Questions | The Importance of Questions and Their Place in the Operationalization of the Concept of "Sociocultural Competence of Labour Migrants in Intercultural Interaction" |
| --- | --- |
| 5. Questions designed to assess the emotional dimension of socio-cultural competence of respondents (7 questions) | The block of questions is aimed at measuring the emotional dimension of sociocultural competence (or "cultural sensitivity"), which includes: the ability to feel comfortable communicating with people of a different race, culture, or belief; empathy and tolerance for cultural differences; non-discrimination and racist behavior; willingness to understand another culture; desire to get acquainted with the best historical and cultural values of the country. Questions were formed in two directions: measuring the tolerance and openness of respondents; measuring the respondents' initiative in establishing contacts with foreign citizens. |
| 6. Questions on respondents' perception of the most important elements of intercultural competence (9 questions) | The block of questions aimed at identifying the list of elements of intercultural competence that respondents consider the most important. It is likely that respondents' perceptions of elements of their intercultural competence may determine their personal priorities (or the priorities that exist in their organization) for areas that need improvement in the first place. |
| 7. Questions aimed at assessing the causes of problem situations (9 questions) | The block of questions is designed to assess the most common causes of misunderstandings when communicating with representatives of other cultures. Respondents were asked to indicate the factors they consider the most common and significant obstacles to effective cooperation with foreign nationals. |
| 8. Questions designed to assess the personal competence that employees should have in a multicultural environment. (9 questions) | The block of questions is aimed at assessing by migrants what personal competence workers should have in a multicultural work environment. Personal competence is assessed on the basis of the following aspects: the ability to present oneself; autonomy; responsibility; initiative; determination; self-confidence; self-motivation; emotional stability and emotional self-control; the ability to adequately assess themselves. |
| 9. Questions to assess migrants' perceptions of the socio-cultural competence that managers should have (9 questions) | This set of statements aims to assess how respondents assess the socio-cultural competence that managers should have in a multicultural work environment: influence on others; conflict management; public speaking skills; engagement in self-education and teaching others; communication and cooperation; team work; group work and tolerance. |
| 10. Questions to find out the ability of respondents to assess the intercultural competence of their colleagues (2 questions) | The block of questions is aimed at assessing the ability of employees to develop and apply assessments of intercultural competence of colleagues. It is clarified how the assessment can be carried out, who should be evaluated, who can be the evaluators, what methods should be used. |

## Appendix B

Questionnaire "Socio-cultural competence of potential labour migrants and migrants who have abroad work experience in the context of migration risks".

Dear respondent, we ask you to answer this anonymous questionnaire, the results of which will be used for scientific purposes and will be published only in statistical form. This study is conducted to assess the work environment and intercultural competence of employees and the need for their development. The intercultural environment consists of people belonging to different nationalities, religions, cultures. Intercultural competence is very important for an employee to be able to communicate successfully and work effectively in such an environment.

I.  Select an answer:

1.  Your gender:

    (a)  woman
    (b)  man

2.  Your age:

    (a)  <25 years
    (b)  >25 to 35 years
    (c)  >35 to 50 years
    (d)  >50 years

3.  Your education:

    (a)  secondary
    (b)  bachelor (vocational, incomplete higher)
    (c)  master's degree (full higher)
    (d)  candidate, doctor of science

4.  Please write:

    country...................................
    oblast....................................
    the city where you work...................

5.  Please indicate which sector you represent:

    (a)  private sector
    (b)  public sector

        (b1)  governing body
        (b2)  health care institute
        (b3)  ultural institutions
        (b4)  educational institutions
        (b5)  other services

    (c)  non-governmental organizationpublic sector
    (d)  other

6.  What is the size of your organization?

    (a)  small (<50 people)
    (b)  middle (from 50 to 250 employees)
    (c)  big (>250 employees)

7.  What is your work experience:

    (a)  less than 1 year
    (b)  1–5 years
    (c)  5 years or more

8.  Your status:

    (a)  student
    (b)  employee
    (c)  specialist
    (d)  head

    (e)  businessman

    (f)   other

II.  Please assess the needs for the development of a multicultural work environment and intercultural competence of employees

    II. 1.  Does your work need to communicate with foreign partners?

        (a)  Yes, several times a quarter on average

        (b)  Yes, once a quarter on average

        (c)  Yes, once a year on average

        (d)  I do not communicate

    II. 2.  Means of communication with foreign partners:

        (a)  Online communication

        (b)  Telephone/fax

        (c)  Correspondence with Ukrposhta

        (d)  Personal meetings

        (e)  Other

    II. 3.  How often do you go to work abroad?

        (a)  On average, once a month

        (b)  Several times a year

        (c)  Once every few years

        (d)  Never

    II. 4.  How often do you go abroad for personal purposes?

        (a)  Weekly

        (b)  Once a month on average

        (c)  Several times a year

        (d)  Less than once a year

        (e)  Never

    II. 5.  The main obstacles to communication with foreign employers:

        (a)  Insufficient awareness

        (b)  Lack of communication skills

        (c)  Disrespect for cultural values

        (d)  Other obstacles

    II. 6.  With citizens of what countries do you communicate most often at work? (You can mark more than one answer)

        (a)  Northern Europe (Denmark, Norway, Sweden and others)

        (b)  Southern Europe (Italy, Greece and others)

        (c)  Eastern Europe (Russia, Ukraine, Belarus and others)

        (d)  Central Europe (Czech Republic, Hungary, Austria and others)

        (e)  Western Europe (France, Germany, Spain, Great Britain and others)

        (f)  The Baltic States (Lithuania, Latvia, Estonia)

        (g)  Far East (China, South Korea, Japan, Philippines and others)

        (h)  Middle East (Iran, Iraq, Syria, Israel, Egypt, Qatar, Lebanon and others)

        (i)  South America (Brazil, Argentina, Colombia, Venezuela and others)

        (j)  North America (Canada, USA and others)

(k) Africa (Libya, Sudan, Nigeria and others)

(l) Australia, New Zealand, Indonesia and others

(m) Others

II. 7. Please rate which elements of intercultural competence are most important in your work environment? (5 means very important, 1 means not important)

| Elements of intercultural competence: | 1 | 2 | 3 | 4 | 5 |
|---|---|---|---|---|---|
| Knowledge of the international protocol of work | 1 | 2 | 3 | 4 | 5 |
| Knowledge of a foreign language (s) | 1 | 2 | 3 | 4 | 5 |
| Understanding politeness | 1 | 2 | 3 | 4 | 5 |
| Knowledge of the main historical and social phenomena of another country | 1 | 2 | 3 | 4 | 5 |
| Flexibility of communication | 1 | 2 | 3 | 4 | 5 |
| Management of new situations | 1 | 2 | 3 | 4 | 5 |
| Learning in test situations | 1 | 2 | 3 | 4 | 5 |
| Tolerance of culture and race differences | 1 | 2 | 3 | 4 | 5 |
| Respect for the values of other cultures | 1 | 2 | 3 | 4 | 5 |
| Other (please specify) | 1 | 2 | 3 | 4 | 5 |

II. 8. How do you stimulate the development of intercultural competence in the workplace?

(a) During empirical courses, seminars, simulation games

(b) Employees must be sent on internships or business trips abroad

(c) Employees learn from each other to stimulate the spread of positive experiences

(d) I do not stimulate the development of intercultural competence, I rely on the responsibility of the employee

(e) I'm not interested

(f) I don't know

II. 9. Do you think the development of intercultural competence could contribute to the success of your career?

(a) Yes, it would improve international career opportunities

(b) Yes, there would be no need to change your job for the better

(c) Yes, career opportunities in my current job would be better

(d) I think that my career does not depend on the level of my intercultural competence

(e) I don't know

(f) I'm not interested

II. 10. Do you think the development of intercultural competence could contribute to the success of your company?

(a) Yes, trust in the organization would be higher

(b) Yes, partnership opportunities with organizations from other countries would be better

(c) Yes, the desire to achieve the goals and results of the organization will be better ensured

(d) I think the success of my organization does not depend on the level of my intercultural competence

(e) I don't know

(f) I'm not interested

II. 11. In your opinion, how will the need for international communication in your organization change over the next five years?

    (a) I think the need will increase (because more foreigners will come, the number of implemented international projects will increase, etc)

    (b) I think the need will remain the same

    (c) I think the need will decrease (because the projects will end, there will be no new initiatives, etc)

    (d) I don't know, I have no idea

II. 12. Please rate yourself according to the statements below (5 means full agreement, 1 means complete disagreement):

| | | | | | |
|---|---|---|---|---|---|
| I can speak a foreign language (s) in professional activities | 1 | 2 | 3 | 4 | 5 |
| I know how to properly introduce myself, exchange greetings, and communicate with foreign citizens during the first acquaintance | 1 | 2 | 3 | 4 | 5 |
| I know about acceptable and unacceptable questions when communicating with foreigners | 1 | 2 | 3 | 4 | 5 |
| I know how to behave at the table in the company of foreigners | 1 | 2 | 3 | 4 | 5 |
| I understand the time rules—where, when and for how much it is allowed to be late | 1 | 2 | 3 | 4 | 5 |
| I understand the appropriate and inappropriate colours of clothing—I know what to wear | 1 | 2 | 3 | 4 | 5 |
| I know about the public order of the countries I visit | 1 | 2 | 3 | 4 | 5 |
| I know about places where smoking is allowed | 1 | 2 | 3 | 4 | 5 |
| I know how to use the public transport system | 1 | 2 | 3 | 4 | 5 |
| I know about the usual amount of tips in the country | 1 | 2 | 3 | 4 | 5 |
| I know about acceptable food times and typical meals; how much and what spirits are suitable for lunch and dinner | 1 | 2 | 3 | 4 | 5 |
| I know how to write a business letter to foreigners properly | 1 | 2 | 3 | 4 | 5 |
| I know the right style of clothing for business meetings | 1 | 2 | 3 | 4 | 5 |
| I know how to greet foreign citizens and business partners for the holidays and when business gifts are suitable | 1 | 2 | 3 | 4 | 5 |
| I know about the levels of subordination that are accepted in the country | 1 | 2 | 3 | 4 | 5 |
| When communicating with foreigners, I know: | | | | | |
| Prohibitions | 1 | 2 | 3 | 4 | 5 |
| Traditions | 1 | 2 | 3 | 4 | 5 |
| Expressions of courtesy | 1 | 2 | 3 | 4 | 5 |
| I can compare some nuances of my culture and smooth out similarities/differences | 1 | 2 | 3 | 4 | 5 |
| Holidays that are important to their country | 1 | 2 | 3 | 4 | 5 |
| What I know about my country | 1 | 2 | 3 | 4 | 5 |
| What I know about the political situation in the country | 1 | 2 | 3 | 4 | 5 |

| | | | | | |
|---|---|---|---|---|---|
| I understand how contacts between foreigners take place and how to communicate | 1 | 2 | 3 | 4 | 5 |
| How they relate to the relationship between managers and subordinates | 1 | 2 | 3 | 4 | 5 |
| How they relate to the relationship between men and women | 1 | 2 | 3 | 4 | 5 |
| How they relate to the relationship between young and old | 1 | 2 | 3 | 4 | 5 |
| When communicating with foreigners, I can say: | | | | | |
| How do they usually solve their problems | 1 | 2 | 3 | 4 | 5 |
| What are the features of their communication and negotiations | 1 | 2 | 3 | 4 | 5 |
| Which method of expressing emotions is acceptable | 1 | 2 | 3 | 4 | 5 |
| When addressing new acquaintances, I do not give any priority to the representative of my nationality | 1 | 2 | 3 | 4 | 5 |
| When communicating with a foreigner, I am interested in the culture, customs, interests of his/her country | 1 | 2 | 3 | 4 | 5 |
| When communicating with a foreigner, I am happy to talk about my country, I want to introduce him/her to the culture of my country | 1 | 2 | 3 | 4 | 5 |
| I try to find out what way of communication is acceptable for a foreigner and try to avoid unacceptable behavior that may offend him/her | 1 | 2 | 3 | 4 | 5 |
| I understand and will tolerate cultural differences | 1 | 2 | 3 | 4 | 5 |
| Race differences | 1 | 2 | 3 | 4 | 5 |
| Religious diversity | 1 | 2 | 3 | 4 | 5 |
| Statements to assess the skills of the respondent | | | | | |
| Communication with foreigners is not a stressor for me and does not cause self-doubt | 1 | 2 | 3 | 4 | 5 |
| I know how to behave in unexpected and new situations that have arisen because of cultural diversity | 1 | 2 | 3 | 4 | 5 |
| I am flexible when I communicate with foreign citizens | 1 | 2 | 3 | 4 | 5 |
| I observe and understand what I learned while communicating with foreign citizens | 1 | 2 | 3 | 4 | 5 |
| In case of conflicts or misunderstandings due to cultural differences, I know how to resolve them properly | 1 | 2 | 3 | 4 | 5 |
| Statements to clarify the problems that most often arise when communicating with foreigners | | | | | |
| Different temperament | 1 | 2 | 3 | 4 | 5 |
| Language | 1 | 2 | 3 | 4 | 5 |
| Different perceptions of communication between managers and subordinates | 1 | 2 | 3 | 4 | 5 |
| Different styles of informal communication | 1 | 2 | 3 | 4 | 5 |
| Different religions | 1 | 2 | 3 | 4 | 5 |

| | | | | | |
|---|---|---|---|---|---|
| Various prohibitions | 1 | 2 | 3 | 4 | 5 |
| Different methods of decision making | 1 | 2 | 3 | 4 | 5 |
| Ignorance of the culture of foreigners | 1 | 2 | 3 | 4 | 5 |
| Employees are not interested in communicating with foreigners | 1 | 2 | 3 | 4 | 5 |

## Appendix C

Statistical analysis of selected data of the author's empirical research.

**Table A2.** Reliability indicators for the Cronbach's Alpha coefficient.

| | Scale | Scale deviation | Corrected Position-General Correlation | Cronbach's Alpha |
|---|---|---|---|---|
| Your gender: | 288.93 | 1973.85 | −0.1 | 0.85 |
| Your age: | 288.4 | 1968.35 | 0 | 0.85 |
| Your education: | 288.16 | 1977.33 | −0.1 | 0.85 |
| Region | 288.4 | 1937.12 | 0.12 | 0.85 |
| Oblast | 288.5 | 1893.17 | 0.34 | 0.85 |
| The city where the respondent works | 283.52 | 1585.45 | 0.41 | 0.87 |
| IP-rated city | 282.84 | 1707.12 | 0.2 | 0.89 |
| Please indicate which sector you represent: | 288.69 | 1965.55 | 0.06 | 0.85 |
| What is the size of your organization? | 288.95 | 1967.35 | 0.03 | 0.85 |
| What is your work experience? | 288.43 | 1969.3 | 0 | 0.85 |
| Your status | 288 | 1986.88 | −0.15 | 0.85 |
| How often do you go to work abroad? | 287.67 | 1979.7 | −0.12 | 0.85 |
| How often do you go abroad for personal purposes? | 287 | 1969.02 | 0 | 0.85 |
| The main obstacles in communication with foreign partners | 287.83 | 1962.81 | 0.02 | 0.85 |
| Knowledge of international labour protocol | 287.22 | 1950.81 | 0.19 | 0.85 |
| Knowledge of a foreign language (c) | 286.36 | 1944.73 | 0.28 | 0.85 |
| Understanding politeness | 286.43 | 1956.28 | 0.14 | 0.85 |
| Knowledge of the main historical and social phenomena of another country | 287.09 | 1945.59 | 0.21 | 0.85 |
| Flexibility of communication | 286.81 | 1944.05 | 0.27 | 0.85 |
| Management of new situations | 287.03 | 1952.95 | 0.17 | 0.85 |
| Learning in test situations | 286.88 | 1949.3 | 0.2 | 0.85 |
| Tolerance of culture and race differences | 286.62 | 1952.27 | 0.2 | 0.85 |

**Table A2.** *Cont.*

| | Scale | Scale deviation | Corrected Position-General Correlation | Cronbach's Alpha |
|---|---|---|---|---|
| Respect for the values of other cultures | 286.53 | 1947.41 | 0.29 | 0.85 |
| Other | 287.47 | 1938.18 | 0.21 | 0.85 |
| Do you think the development of intercultural competence could contribute to the success of your career? | 287.98 | 1984.47 | −0.12 | 0.85 |
| Do you think the development of intercultural competence could contribute to the success of your organization? | 287.83 | 1991.23 | −0.17 | 0.85 |
| In your opinion, how will the need for international communication in your organization change over the next five years? | 288.79 | 1976.45 | −0.08 | 0.85 |
| I can communicate in a foreign language (s) in my professional activity | 286.97 | 1923.3 | 0.46 | 0.85 |
| I know how to properly introduce myself, exchange greetings and communicate with foreign citizens during the first acquaintance | 286.72 | 1907.54 | 0.62 | 0.85 |
| I know about acceptable and unacceptable questions when communicating with foreigners | 286.9 | 1915.5 | 0.59 | 0.85 |
| I know how to behave at the table | 286.6 | 1911.79 | 0.56 | 0.85 |
| I understand the time rules: where, when and for how much it is allowed to be late | 286.59 | 1920.63 | 0.47 | 0.85 |
| I understand the appropriate and inappropriate colours of clothing to wear | 286.76 | 1908.43 | 0.56 | 0.85 |
| I know about the public order of the countries I visit | 286.81 | 1907.67 | 0.62 | 0.85 |
| I know about places where smoking is allowed | 286.76 | 1917.59 | 0.53 | 0.85 |
| I know how to use the public transport system | 286.81 | 1908.72 | 0.67 | 0.85 |
| I know about the usual amount of tips in the country | 287.05 | 1914.82 | 0.59 | 0.85 |
| I know that there is a reasonable time for food and typical dishes, how much and what spirits are suitable for lunch and dinner | 286.93 | 1911.68 | 0.54 | 0.85 |

**Table A2.** *Cont.*

| | Scale | Scale deviation | Corrected Position-General Correlation | Cronbach's Alpha |
|---|---|---|---|---|
| I know how to write a business letter to foreigners properly | 287.07 | 1900.52 | 0.64 | 0.85 |
| I know the right style of clothing for business meetings | 286.67 | 1914.4 | 0.55 | 0.85 |
| I know how to greet foreign citizens and business partners for the holidays and when business gifts are appropriate | 286.93 | 1909.78 | 0.65 | 0.85 |
| I know about the levels of subordination that are accepted in the country | 286.97 | 1905.26 | 0.65 | 0.85 |
| I know what their prohibitions are | 287.02 | 1913.7 | 0.51 | 0.85 |
| I know what their traditions are | 286.97 | 1914.98 | 0.54 | 0.85 |
| I know what their courtesies are | 286.72 | 1917.4 | 0.5 | 0.85 |
| I can compare some nuances of my culture and smooth out the differences | 286.88 | 1905.62 | 0.61 | 0.85 |
| I can name holidays that are important for their country | 287 | 1904.28 | 0.6 | 0.85 |
| I can name what I know about my country | 286.4 | 1924.77 | 0.57 | 0.85 |
| I can name what I know about the political situation in the country | 286.57 | 1917.55 | 0.55 | 0.85 |
| I understand how they relate to the relationship between managers and subordinates | 286.88 | 1912.07 | 0.62 | 0.85 |
| I understand how they relate to the relationship between men and women | 286.72 | 1922.31 | 0.53 | 0.85 |
| I understand how they relate to the relationship between young and old | 286.64 | 1916.02 | 0.68 | 0.85 |
| I understand how they usually solve their problems | 286.78 | 1910.84 | 0.7 | 0.85 |
| I understand the peculiarities of their communication and negotiations | 286.86 | 1916.23 | 0.57 | 0.85 |
| I understand which method of expressing emotions is acceptable | 286.91 | 1907.38 | 0.58 | 0.85 |
| When addressing new acquaintances, I do not give any priority to a representative of my nationality | 286.79 | 1916.73 | 0.55 | 0.85 |

**Table A2.** *Cont.*

|  | Scale | Scale deviation | Corrected Position-General Correlation | Cronbach's Alpha |
|---|---|---|---|---|
| When communicating with a foreigner, I am interested in the culture, customs, interests of his/her country | 286.76 | 1923.13 | 0.54 | 0.85 |
| When communicating with a foreigner, I am happy to talk about my country, I want to introduce him/her to the culture of my country | 286.62 | 1914.13 | 0.66 | 0.85 |
| I try to find out what way of communication is acceptable for a foreigner and try to avoid unacceptable behaviour that may offend him/her | 286.5 | 1931.55 | 0.48 | 0.85 |
| I understand and will tolerate cultural differences | 286.6 | 1914.1 | 0.63 | 0.85 |
| Race differences | 286.84 | 1910.1 | 0.56 | 0.85 |
| Religious diversity | 286.71 | 1919.75 | 0.49 | 0.85 |
| Communication with foreigners is not stressful for me | 286.69 | 1927.83 | 0.42 | 0.85 |
| I know how to behave in unexpected and new situations that have arisen because of cultural diversity | 286.84 | 1914.8 | 0.58 | 0.85 |
| I am flexible when I communicate with foreign citizens | 286.9 | 1923.6 | 0.47 | 0.85 |
| I observe and understand what I learned while communicating with foreign citizens | 286.83 | 1913.09 | 0.58 | 0.85 |
| In case of conflicts or misunderstandings due to cultural differences, I know how to resolve them properly | 286.95 | 1912.33 | 0.61 | 0.85 |
| Different temperament | 287.57 | 1955.34 | 0.08 | 0.85 |
| Language | 287.14 | 1942.82 | 0.17 | 0.85 |
| Different perceptions of communication between managers and subordinates | 287.33 | 1937.91 | 0.22 | 0.85 |
| Different styles of informal communication | 287.17 | 1940.71 | 0.2 | 0.85 |
| Different religions | 287.4 | 1955.16 | 0.09 | 0.85 |
| Various prohibitions | 287.34 | 1949.49 | 0.13 | 0.85 |

**Table A2.** *Cont.*

|  | Scale | Scale deviation | Corrected Position-General Correlation | Cronbach's Alpha |
| --- | --- | --- | --- | --- |
| Different methods of decision making | 287.48 | 1959.24 | 0.05 | 0.85 |
| Ignorance of the culture of foreigners | 287.19 | 1953.17 | 0.1 | 0.85 |
| Employees are not interested in foreigners | 287.47 | 1947.73 | 0.13 | 0.85 |

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
