# Peer review of "Social Risks of International Labour Migration in the Context of Global Challenges"

_jrfm, doi:10.3390/jrfm13090197_

Round 1

Reviewer 1 Report

Dear authors, the manuscript requires extensive changes related to the scientific soundness of the analysis. Also the language needs to be improved. I think the work needs to be reconsidered after major changes to the text. Please find my comments below.

As a general comment, the manuscript in its current status, does not reflect the purpose of the study as outlined by the authors: “The purpose of the study is to determine conditions of migration social risk formation, ways to overcome them and possible consequences for experienced and potential migrants in foreign cultural environment.” Furthermore, it seems that many statements in the text are based on claims, rather than scientific literature or evidence, please add scientific basis to what you state. More details in my comments below.

Abstract: it needs to be rewritten, language is not clear, the methodology is not outlined, results are presented in a rather confusing way.

Introduction: overall references are missing in this section from line 30 to 60. References to risk theory are missing; also, it is crucial to provide a definition of “labour migrants” and “experienced migrants”, this is missing in the text. Please provide examples of repulsion and attraction factors (line 85).

Line 32-35: This must be justified with scientific literature and previous studies, otherwise these claims must be removed. E.g. in many countries, migrants do jobs that locals are not willing to do, therefore migrants would not affect unemployment in host countries in this sense.

Line 38-42: please add references or revise the paragraph.

Line 55-60: the socio-political sphere is not discussed.

Line 70-80: main concepts of methodology and outcomes of other works need to be briefly outlined, not just simply mentioned as a long list of publications. Especially the relevance of these studies for the current paper must be discussed

Line 89: “(…) makes it possible to consider international labour migration as a vital strategy”, a vital strategy for what? This is unclear.

Materials and methods: this section needs to be expanded; references added; methods, databases explained. more information on the conducted survey also needs to be given, e.g. selection of migrants, which is the host country under study, content of the questionnaires.

Results: it seems that the migrant knowledge of the foreign language of host countries is almost the only topic addressed in the study. Topics listed in line 249-252 should be further developed in the section to give a comprehensive overview of the survey.

Line 116-118: What is the scientific basis to say that “social risks of international labour migrants are shaped by their own choices”? The host country and its political approach to the topic have typically a huge influence on how this migration is handled and which opportunities are given to migrants.

Line 121-123: again, is this based on scientific findings or is it just a claim? Is this a conclusion derived from literature?

Line 123-125:  I think that migrants do not leave their countries of origin because they feel like to, but rather due to a willingness to improve their conditions in comparison to the country of origin. In this fight for a better working condition and life, it is clear that all other risks in other host country are less relevant for migrants. Is this point discussed anywhere in the text? Also, social risks are very different depending on workers´ qualifications and the type of labour migrants analysed in a study.

Line 126-128: please rephrase, this sounds contradicting.

Line 196-198: “(…) similarity shows up in refusal to recognize poor command of the host country language.” Do the authors have previous studies that analyse the real knowledge of the host country language by the same interviewed migrants? Or is the poor command of the host country language something that the authors expected, on which basis?

Line 206-208: these results need to be compared to real data on the language skills of migrants, in order to compare perception (from questionnaires) and reality. It seems that rather the authors assume that language level of migrants should be low or anyway lower than the perception. However, it is not mentioned for how long interviewed migrants have been living or working abroad.

Line 213-214: not clear, please revise

Figure 1: why is the scale in the figure (0-9) different from that proposed (1-5)? Or what is the meaning of the scale?

Discussion: this section does not really discuss results nor the results against the purpose of the study “The purpose of the study is to determine conditions of migration social risk formation, ways to overcome them and possible consequences for experienced and potential migrants in foreign cultural environment.”

Conclusion: the concept of cross-cultural competence is not explained in the text. Little discussion on the social risks of labour migration is mentioned in this section, but nor really presented in the results and discussions sections before.

Author Response

Dear Reviewer,

Many thanks for your time and valuable comments! Attached please find our detailed responses to your comments.

The article was translated into English by a native speaker. If you have comments about the translation of the text, please write in which lines or paragraphs.

Thanks again!
Best regards

Reviewer 2 Report

Journal

JRFM (ISSN 1911-8074)

Manuscript ID

jrfm-872908

Title

Social risks of international labor migration in the context of global challenges

I regret to inform you that I find no scientific validity to this work. I recommend to reject the manuscript based on the following points:

  1. The title does not correspond to the work done. “The purpose of the study is to determine conditions of migration social risk formation, ways to overcome them and possible consequences for experienced and potential migrants in foreign cultural environment”.

  1. In the abstract, statements are made that are not supported by this study. E.g. “ It has been proven that the greatest social danger for labor migrants is the loss of competence and initiative components”.
  2. The introduction does not reflect the state of the art, nor does it highlight the importance of the problem to be studied. Note that all references are between 2017 and 2019.
  3. The references are quite incomplete. Journal names do not appear in some cases. E.g. Pécoud, A. What do we know about the International Organization for Migration?, 2018
  4. riskology? Please revise this term.
  5. Materials and Methods.

Line 131 A questionnaire was developed for the empirical phase of the study. Where are the questions? The survey is not defined.

  1. There is not a formal statistical analysis (ANOVA) that support these results.
  2. line 134 The joint part of the questionnaire also contained a 5-point scale, where "5" points mean "very important" and "1" not at all important. Figure 1. Distribution of responses of experienced and potential labor migrants regarding the ability to use a foreign language in their professional activity, % (author’s research data) (Is the scale from 0 to 9?).
  3. In the discussion section, it does not compare with previous works, so it is not a proper discussion

In short, the survey carried out has not been defined, the data obtained are not provided, a statistical study has not been carried out, and therefore the conclusions are not scientifically supported

Author Response

(The authors gave the same response as above.)

Round 2

Reviewer 2 Report

The authors have significantly improved this new version. In fact it doesn't look the same. All my suggestions have been taken into account. However, I think the state of the art is still not up to date. This aspect can be further improved.

Minor points:

Figures 1 and 2 are not clear, another way of representation must be found, also the decimal point of the figures is not correct (it must be a point and not a comma)

Author Response

Dear Reviewer.

The authors are sincerely grateful for the suggestions and recommendations provided by the reviewer. The reviewer aptly drew attention to the possibility of improving the technical component of the research. We fully agree with this suggestion. This will be taken into account in further research.

Minor points:

Figures 1 and 2 are not clear, another way of representation must be found, also the decimal point of the figures is not correct (it must be a point and not a comma)

Response: The authors corrected the commas to dots in Figures 1 and 2.

Lines 500, 514, 640.

Round 3

Reviewer 2 Report

The authors have improved their manuscript according to my suggestions. For my side it can be accepted for publication.

Author Response

Thank you.